# Dermoscopy for melanoma detection and triage in primary care: a systematic review

OT Jones,[1] LC Jurascheck,[2] MA van Melle,[1] S Hickman,[3] NP Burrows,[4] PN Hall,[5] J Emery,[1,6] FM Walter[1,6]

[1]Department of Public Health and Primary Care, University of Cambridge, Cambridge, UK
[2]University of Cambridge School of Clinical Medicine, Cambridge, UK
[3]Norfolk and Norwich University Hospitals NHS Foundation Trust, Norwich, UK
[4]Addenbrooke's Hospital Department of Dermatology, Cambridge University Hospitals NHS Foundation Trust, Cambridge, UK
[5]Addenbrooke's Hospital, Cambridge University Hospitals NHS Foundation Trust, Cambridge, UK
[6]General Practice and Primary Care Academic Centre, University of Melbourne, Carlton, Victoria, Australia

**Correspondence to**
Dr OT Jones;
otj24@medschl.cam.ac.uk

## ABSTRACT

**Objective** Most skin lesions first present in primary care, where distinguishing rare melanomas from benign lesions can be challenging. Dermoscopy improves diagnostic accuracy among specialists and is promoted for use by primary care physicians (PCPs). However, when used by untrained clinicians, accuracy may be no better than visual inspection. This study aimed to undertake a systematic review of literature reporting use of dermoscopy to triage suspicious skin lesions in primary care settings, and challenges for implementation.

**Design** A systematic literature review and narrative synthesis.

**Data sources** We searched MEDLINE, Cochrane Central, EMBASE, Cumulative Index to Nursing and Allied Health Literature, and SCOPUS bibliographic databases from 1 January 1990 to 31 December 2017, without language restrictions.

**Inclusion criteria** Studies including assessment of dermoscopy accuracy, acceptability to patients and PCPs, training requirements, and cost-effectiveness of dermoscopy modes in primary care, including trials, diagnostic accuracy and acceptability studies.

**Results** 23 studies met the review criteria, representing 49 769 lesions and 3708 PCPs, all from high-income countries. There was a paucity of studies set truly in primary care and the outcomes measured were diverse. The heterogeneity therefore made meta-analysis unfeasible; the data were synthesised through narrative review. Dermoscopy, with appropriate training, was associated with improved diagnostic accuracy for melanoma and benign lesions, and reduced unnecessary excisions and referrals. Teledermoscopy-based referral systems improved triage accuracy. Only three studies examined cost-effectiveness; hence, there was insufficient evidence to draw conclusions. Costs, training and time requirements were considered important implementation barriers. Patient satisfaction was seldom assessed. Computer-aided dermoscopy and other technological advances have not yet been tested in primary care.

**Conclusions** Dermoscopy could help PCPs triage suspicious lesions for biopsy, urgent referral or reassurance. However, it will be important to establish further evidence on minimum training requirements to reach competence, as well as the cost-effectiveness and patient acceptability of implementing dermoscopy in primary care.

**Trial registration number** CRD42018091395.

## Strengths and limitations of this study

► This study systematically reviews the published evidence for dermoscopy use by primary care physicians in primary care settings, including studies of acceptability and cost-effectiveness, as well as diagnostic accuracy studies.
► The use of a broad search strategy across multiple databases enabled us to identify 23 studies whose findings examine dermoscopy use in primary care clinical practice.
► The included studies were of varying quality.
► Due to the heterogeneity of the included papers, we were not able to undertake any meta-analysis; instead, we performed a narrative synthesis.

## INTRODUCTION

Worldwide malignant melanoma is the 15th most common cancer.[1] Melanoma has one of the fastest rising incidence rates of any cancer, and among white populations incidence has quadrupled over the last 30 years. In the UK this is projected to rise by a further 7% between 2014 and 2035, reflecting increasing exposure to the main risk factor, ultraviolet radiation.[2] There were nearly 300 000 new cases of melanoma worldwide in 2018.[1]

Primary care (the first point of contact for patients in the healthcare system, usually community-based) can play an important role in improving outcomes for patients with melanoma. More accurate triage of suspicious pigmented skin lesions could lead to more prompt diagnosis of melanoma at earlier stages and improved outcomes, and reduce unnecessary biopsies or referrals. Most people diagnosed with cancer first present in primary care,[3] where primary care physicians (PCPs) need to distinguish rare melanomas from common benign lesions using clinical history taking and visual inspection, aided by checklists such as the 7-point checklist as recommended in the 2015 National Institute for Health and Care Excellence guidelines

for suspected cancer.[4] Various technologies may also have a role in assisting triage of suspicious skin lesions, including mobile phone applications,[5] reflectance confocal microscopy,[6] optical coherence tomography,[7] computer-aided diagnosis,[8] high-frequency ultrasound[9] and dermoscopy.[10]

Dermoscopy (also referred to as dermatoscopy or epiluminescence microscopy) is a non-invasive technique using a hand-held magnifier and incident light, which may be polarised to reduce reflection, to reveal subsurface structures. Dermoscopy performed by trained specialists is more sensitive and specific in classifying skin lesions than clinical examination with the naked eye alone.[4 11] Dermatologists and some international guidelines recommend PCPs use dermoscopy[12]; however, when used by untrained or less experienced clinicians, accuracy can be no better than inspection alone,[13] and there is a danger of increased excisions, over-referral or false reassurance. It takes time to train clinicians to use dermoscopy, and PCP training dropout rates have been shown to be high.[14 15] For these reasons dermoscopy is not currently recommended for use by PCPs in the UK,[4] although it is used routinely by PCPs in Australia,[16] which has the highest incidence of melanoma worldwide. Some digital dermoscopy devices exist, a few of which incorporate computer-aided diagnosis, but they are expensive, and while showing better sensitivity even in expert hands many have lower specificity than clinicians alone.[17] However, recent research suggests computer-aided diagnostic tools have the potential to exceed the diagnostic performance of dermatologists.[18]

A Cochrane review of dermoscopy has recently been published and examines the diagnostic accuracy of dermoscopy, with and without visual inspection, for the detection of cutaneous invasive melanoma and intraepidermal melanocytic variants in adults.[19] Our systematic review has a broader aim, focusing on the first presentation of suspicious skin lesions in primary care and whether dermoscopy and dermoscopy-related technologies, with suitable training, can be used accurately and effectively to triage suspicious skin lesions at this point in the healthcare pathway. We considered various types of dermoscopy technologies, including hand-held dermoscopy, computer-aided/digital dermoscopy devices and novel teledermoscopy approaches (ie, referral using electronic dermoscopy images or video). In addition to data on the diagnostic accuracy of dermoscopy, we looked for data on the practical challenges to implementing dermoscopy in primary care, including utility, acceptability to patients and PCPs, training requirements, and cost-effectiveness.

## METHODS

This systematic review was conducted in accordance with the Preferred Reporting Items for Systematic Reviews and Meta-Analyses (PRISMA) guidelines,[20] and the protocol was registered with PROSPERO prior to conducting the review.[21] All aspects of the protocol were reviewed by senior faculty from the CanTest Collaborative (www.cantest.org).

We searched the MEDLINE, EMBASE, Cochrane Central, Cumulative Index to Nursing and Allied Health Literature, and SCOPUS databases using keywords related to dermoscopy, melanoma and primary care, without language restrictions, from 1 January 1990 to 31 December 2017. We also manually searched the reference lists of included studies. We included all types of study design as we anticipated that there would be few relevant randomised controlled trials (RCTs) or diagnostic accuracy studies performed in primary care, and we aimed to find additional qualitative evidence on barriers to the use of dermoscopy which may be found in non-RCT study designs. We chose to start the search from 1990 as this was when the earliest dermoscopy-related research emerged. We considered published evidence from any international healthcare system and whether it could be interpreted and applied to primary care settings, including the extent to which data collected from specialist clinic settings could be applied to the lower-prevalence primary care population.

We included all studies which provide evidence around test accuracy, utility, acceptability to patients and PCPs, training requirements, and cost-effectiveness of dermoscopy modes in primary care, including trials, diagnostic accuracy and acceptability studies. As our interest was in the use of dermoscopy by generalist clinicians, we included all studies reporting PCP use of dermoscopy; studies of secondary care physicians who were not trained in dermoscopy were assessed for the applicability of their study to answer the research question. We excluded studies that were based in any clinical setting other than at the first assessment of suspicious skin lesions, and any studies that were not considered primary studies.

Following duplicate removal, one author (OJ) screened titles and abstracts to identify studies which fitted the inclusion criteria. Of the titles and abstracts 10% were checked by two other authors (LJ and SH), and interassessor reliability was excellent, with disagreement for only 1 out of the 100 papers checked. Any disagreements were discussed by the core research team (OJ, LJ, SH, FMW) and a consensus reached. At least two reviewers (OJ, LJ, SH, MvM, FMW) independently assessed each full-text article for the possibility of inclusion in the review. Any disagreements were resolved by consensus-based discussion.

Data extraction was undertaken by two reviewers independently (OJ, LJ, FMW) and summarised using descriptive tables, discussion and consensus. We chose to extract only reported outcomes from the included papers, without calculating further quantitative measures of diagnostic accuracy from their data, unless already reported. Due to the heterogeneity of the included papers, we were not able to undertake any meta-analysis; instead, we chose to perform a narrative synthesis.

Risk-of-bias assessment was undertaken for each full-text paper by two independent researchers (OJ, LJ) using

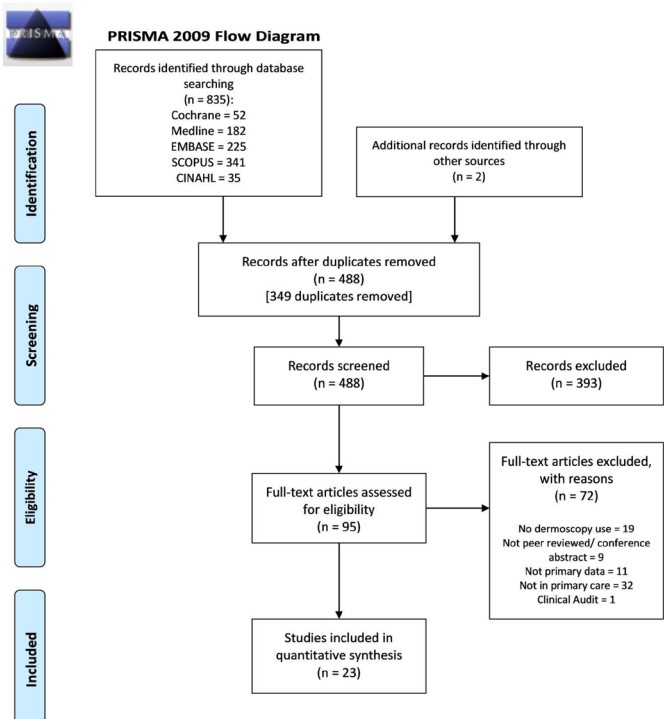

**Figure 1** PRISMA flow diagram for the studies included in the review. CINAHL, Cumulative Index to Nursing and Allied Health Literature; PRISMA, Preferred Reporting Items for Systematic Reviews and Meta-Analyses.

the Joanna Briggs Institute (JBI) critical appraisal tools.[22] These tools incorporate various critical assessments for different study designs, including patient selection, randomisation, data collection and analysis. As assessments for different study designs had varying denominators, the score was converted to a percentage and classified as high, medium and low risk to aid clarity of presentation and interpretation. Although the studies demonstrated a wide range in quality, no studies were excluded based on their risk-of-bias assessment. Full details of our review question, search strategy, inclusion/exclusion criteria, methodology for data extraction, risk-of-bias assessment and outcomes extraction are described in online supplementary appendices 1 and 2, as well as a full list of excluded studies (online supplementary appendix 3).

### Patient and public involvement

Our long-standing collaborator, Mrs Margaret Johnson, is a patient advocate. She commented regularly on the study from its conception, including aspects of the research question, outcome measures and study design. There was no patient recruitment required for this study. The results will be disseminated to patient advocates, groups and relevant charities.

### RESULTS

Figure 1 shows the study PRISMA diagram. There were 837 studies identified, of which 349 were duplicates. Ninety-five articles underwent full-text review and 23 met the inclusion criteria.[14 23–44] These 23 articles reported data relating to 49 769 lesions and 3708 PCPs.

Table 1 provides a summary of the study characteristics for included studies. We included three RCTs, two sequential intervention trials (SIT), nine diagnostic accuracy studies, two cohort studies, two case series, one case–control study and four PCP surveys. Table 1 also visually summarises the practitioner and patient populations reported in the studies and highlights the paucity of studies reporting PCPs using dermoscopy with primary care patients (5 out of 16). Studies of teledermoscopy-based referral systems were more frequently set in primary care, with six out of seven studies involving primary care clinicians and primary care patients. Overall, 16 of the 23 papers reported studies of PCPs, but only 11 papers reported studies involving primary care patients.

Table 2 summarises the outcome measures of each included study, grouped into accuracy and reliability outcomes and implementation outcomes, and shows the heterogeneous nature of the reported outcomes. The accuracy and reliability outcomes were diverse; 12 papers reported sensitivity and specificity, 8 reported diagnostic accuracy or area under the curve, 5 reported positive and negative predictive values, 14 reported the proportion of correct decisions, 4 reported the number needed to excise, and 5 reported the biopsy rate. The implementation outcomes were less numerous but also quite diverse: 4 papers reported on PCP opinions, 3 performed cost-effective analyses, 2 looked at response times for teledermoscopy services, 2 looked at image quality for teledermoscopy, and 1 assessed patient satisfaction.

Risk-of-bias outcomes from the JBI critical appraisal tools are included in table 2, demonstrating a wide range in quality across the studies. No studies were excluded based on the risk-of-bias assessment.

Tables 3 summarises the diagnostic accuracy results, with the studies grouped into RCTs and SITs, non-RCT diagnostic accuracy studies, and survey studies. Among the RCTs and SITs, Argenziano et al,[23] Koelink et al,[24] Rosendahl et al[43] and Menzies et al[14] found that dermoscopy reduced the number needed to excise to diagnose a melanoma. Ferrándiz et al[26] evaluated the impact of adding dermoscopic images to the standard teledermatology referral system and found that it improved accuracy and confidence in diagnosing skin lesions.

Most of the studies were non-RCT diagnostic accuracy studies. These showed increased diagnostic accuracy with the use of dermoscopy in primary care[28 33 35 43 44] or in teledermoscopy-based referral systems.[34–37] Some studies suggested this was due to improved ability to identify benign lesions when using a dermatoscope.[27 31 32 44] All studies that assessed the effect of training found that it improved diagnostic accuracy compared with minimal or no training.[25 29–33 44] There was evidence that use of dermoscopy without training displayed similar diagnostic accuracy to naked-eye examination.[33] Menzies et al[29] showed that a dermoscopy-related technology, SolarScan,

 

**Table 1** Study demographics: patient and practitioner populations

| Study details | Location | Study type | Practitioners | | | | Patients | | | | | | Control group | Intervention group | Gender of patients (except where indicated) (F/M) (%) | Age of patients (except where indicated) (years), mean (range) |
|---|---|---|---|---|---|---|---|---|---|---|---|---|---|---|---|---|
| | | | | | | | Studies with patients from: | | | Studies using images from: | | | | | | |
| | | | PCPs | SCPs | PCPs and SCPs | DPCSCCs | Primary care | Secondary care | DPCSCCs | Primary care | Secondary care | DPCSCCs | | | | |
| Dermoscopy papers | | | | | | | | | | | | | | | | |
| Ahmadi et al[27] | Maastricht/ Limburg, The Netherlands | Case series | ■ | | | | ■ | | | | | | None | Patients from 3 primary care practices | F=57.8 | 54.7 (60–79) |
| Argenziano et al[23] | Barcelona, Spain; Naples, Italy | RCT | ■ | | | | ■ | | | | | | Naked eye | Dermoscopy | C: F=62.4 I: F=62.3 | C: 40 (2–90) I: 41 (3–94) |
| Bourne et al[28] | Brisbane, Australia | DA study | | | | | | | | | | ■ | Clinical assessment and algorithms | BLINCK algorithm | F=52.2 | 58 (30–60) |
| Chappuis et al[38] | 4 regions of France | Survey | ■ | | | | | | | | | | None | PCPs in France | GPs: F=42.4 | GPs: <30=8 30–50=169 >50=246 |
| Koelink et al[24] | Groningen, The Netherlands | RCT | ■ | | | | ■ | | | | | | Naked eye | Dermoscopy | C: F=61.6 I: F=68.2 | C: 54.7 I: 53.2 |
| Menzies et al[29] | USA, Germany and Australia | DA study | | | ■ | | | | | | | ■ | Independent clinicians | SolarScan assessment | NR | NR |
| Menzies et al[14] | Perth, Australia | SIT | | * | ■ | | ■ | | | | | | PCP decision before intervention | Outcome after dermoscopy and SDDI | NR | NR |
| Morris et al[39] | Florida, USA | Survey | ■ | | | | | | | | | | None | PCPs | Clinicians: F=41.6 | Clinicians: median 40–49 years |
| Morris et al[40] | Florida, USA | Survey | | | ■ | | | | | | | | None | Practising physicians | Clinicians: F=34.7 | Clinicians: NR |
| Pagnanelli et al[30] | Rome, Italy | DA study | | | | | | | | ■ | ■ | | Pretraining | Post-training | NR | NR |
| Rogers et al[31] | New York, USA | DA study | | | | | | | | | ■ | | Histology/ expert opinion | Clinicians using 3 algorithms | F=53.3 | Median 31–40 years |
| Rogers et al[32] | New York, USA | DA study | | | ■ | | | | | | ■ | | Histology/ expert opinion | Clinicians using 3 algorithms | F=53.3 | Median 31–40 years |
| Rosendahl et al[43] | Queensland, Australia | SIT | † | | | | | | | | | ■ | Naked eye | Dermoscopy images | F=32.6 | 57 SD: 17 years |
| Rosendahl et al[25] | Australian SCARD database | Cohort study | | | | | | | ■ | | | | Histology diagnosis | PCP decision | NR | NR |
| Secker et al[44] | Leiden, The Netherlands | DA study | | | | | | | | | ■ | | PCPs before education | After education | F=51.8 | 45.2 (28–63) |

Continued

**Table 1** Continued

| Study details | Location | Study type | Practitioners | | | Patients | | | | | | Control group | Intervention group | Gender of patients (except where indicated) (F/M) (%) | Age of patients (except where indicated) (years), mean (range) |
|---|---|---|---|---|---|---|---|---|---|---|---|---|---|---|---|
| | | | | | | Studies with patients from: | | | Studies using images from: | | | | | | |
| | | | PCPs | SCPs | PCPs and SCPs | Primary care | Secondary care | DPCSCCs | Primary care | Secondary care | DPCSCCs | | | | |
| Westerhoff et al[33] | Sydney, Australia | DA study | ☑ | | | | | | | ☑ | | PCP diagnosis | PCPs ± dermoscopy ± education | NR | NR |
| **Teledermoscopy papers** | | | | | | | | | | | | | | | |
| Börve et al[34] | Gothenburg, Sweden | Case-control study | | | | ☑ | | | | | | Paper-based referrals | Teledermoscopy referrals | C: F=57.1 I: F=61.4 | C: 61 (18–97) I: 54 (18–93) |
| Ferrándiz et al[26] | Andalucia, Spain | RCT | | | | ☑ | | | | | | Clinical images | Clinical and dermoscopy images | C: F=52.88 I: F=62.28 | C: 57.33 I: 54.96 |
| Grimaldi et al[35] | Siena, Italy | DA study | | | | ☑ | | | | | | Judgement before dermoscopy | Judgement after dermoscopy | NR | NR |
| Livingstone and Solomon[41] | Ruislip, UK | Case series | | | | ☑ | | | | | | Expert diagnosis and standard costs | Teledermoscopy referrals | NR | NR |
| Moreno-Ramirez et al[36] | Sevilla, Spain | DA study | ☑ | | | ☑ | | | | | | Tele dermatology referrals | Same patients + dermoscopy images | F=70.5 | 38.8 (1–73) |
| Stratton and Loescher[42] | Arizona, USA | Survey | | | ☑ | | | | | | | None | Nurse practitioners | Nurse practitioners: F=92 | Nurse practitioners: 48 |
| van der Heijden et al[37] | Amsterdam, The Netherlands | Cohort study | ☑ | | | ☑ | | | | | | Face-to-face consult ± histology | Teledermoscopy consult (same patients) | F=55 | Median 47 years, 6–84 years |

Coloured boxes denote that practitioners and patients from these populations were included in the corresponding study

*Minimal dermoscopy experience, although secondary care physicians.

†Practitioner population not specified.

BLINCK, Benign, Lonely, Irregular, Nervous, Change, Known clues; C, control group; DA, diagnostic accuracy; DPCSCC, dedicated primary care skin cancer clinic; F, female; GP, General Practitioner; I, intervention group; M, male; NR, not reported; PCP, primary care physician; RCT, randomised controlled trial; SCARD, Skin Cancer Audit Research Database; SCP, secondary care physician; SD, Standard Deviation; SDDI, short-term sequential digital dermoscopy imaging; SIT, sequential intervention trial.

**Table 2** Diversity of reported outcomes and critical appraisal results of the included studies

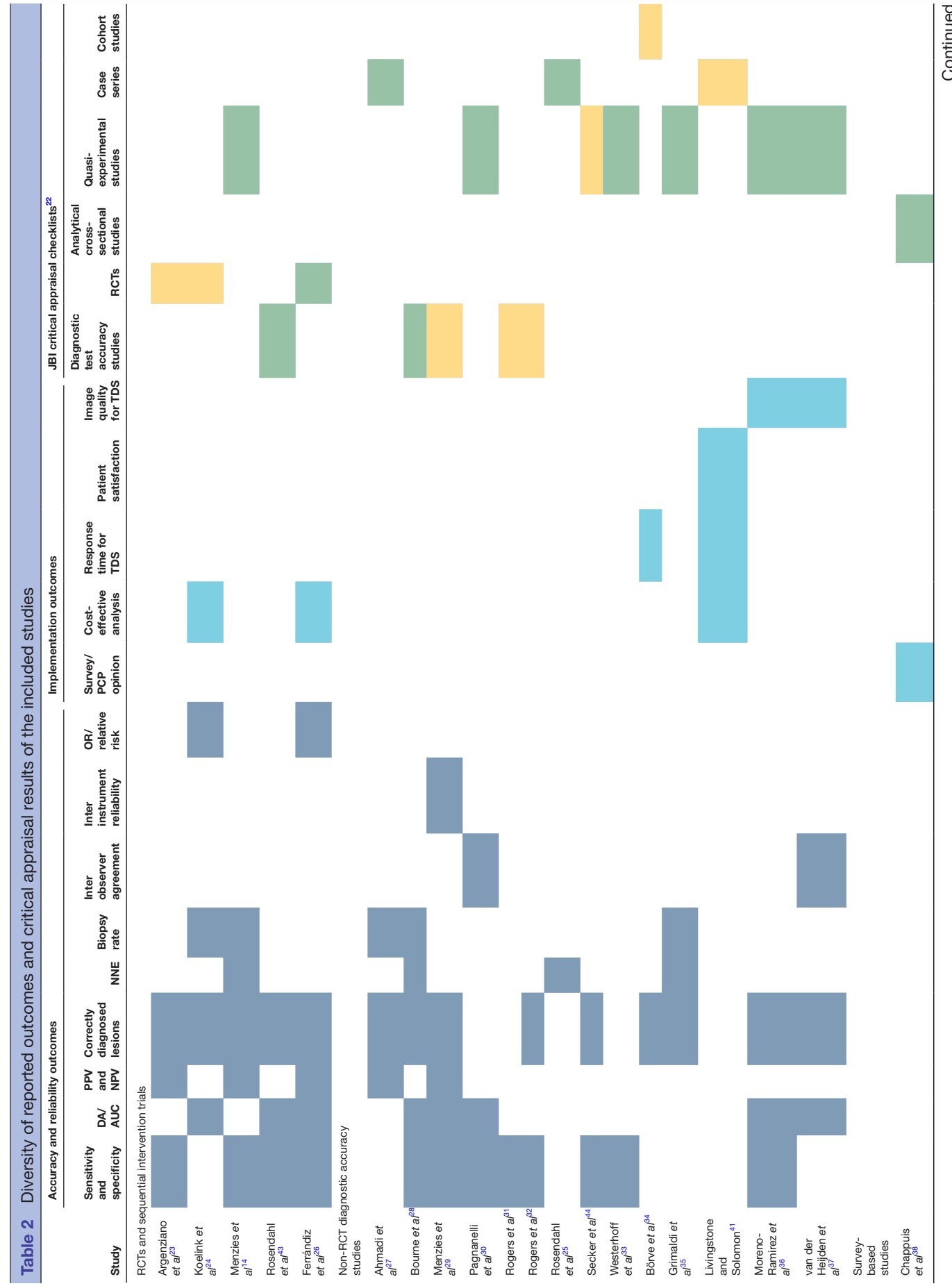

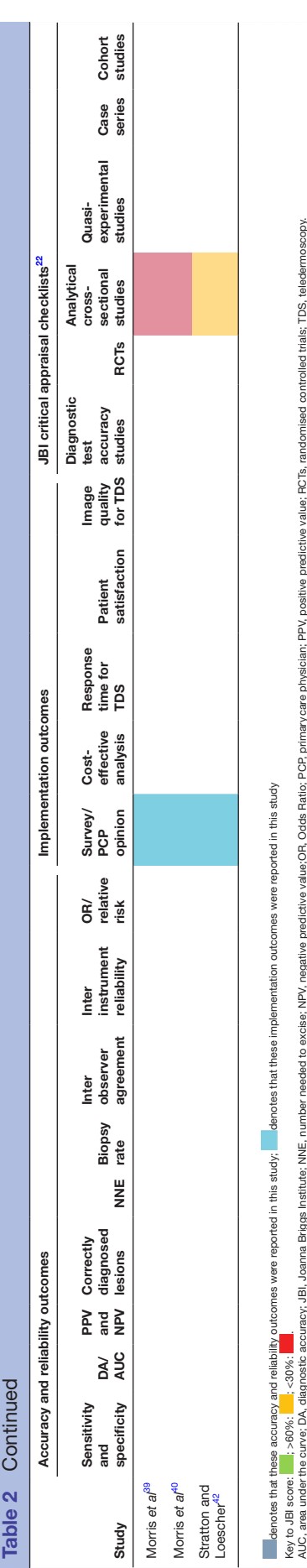

**Table 2** Continued

| | Accuracy and reliability outcomes | | | | | | | | | | Implementation outcomes | | | | | | | JBI critical appraisal checklists[22] | | | | | |
|---|---|---|---|---|---|---|---|---|---|---|---|---|---|---|---|---|---|---|---|---|---|---|---|
| Study | Sensitivity and specificity | DA/ AUC | PPV and NPV | Correctly diagnosed lesions | NNE | Biopsy rate | Inter observer agreement | Inter instrument reliability | OR/ relative risk | Survey/ PCP opinion | Cost-effective analysis | Response time for TDS | Patient satisfaction | Image quality for TDS | Diagnostic test accuracy studies | RCTs | Analytical cross-sectional studies | Quasi-experimental studies | Case series | Cohort studies |
| Morris et al[39] | | | | | | | | | | | | | | | | | | | | |
| Morris et al[40] | | | | | | | | | | | | | | | | | | | | |
| Stratton and Loescher[42] | | | | | | | | | | | | | | | | | | | | |

■ denotes that these accuracy and reliability outcomes were reported in this study; ■ denotes that these implementation outcomes were reported in this study

Key to JBI score: ■ >60%; ■ <30%.

AUC, area under the curve; DA, diagnostic accuracy; JBI, Joanna Briggs Institute; NNE, number needed to excise; NPV, negative predictive value; OR, Odds Ratio; PCP, primary care physician; PPV, positive predictive value; RCTs, randomised controlled trials; TDS, teledermoscopy.

had higher sensitivity than PCPs, although this was a non-significant finding.

Table 4 summarises findings from the studies which investigated barriers and facilitators to implementing dermoscopy in primary care. Training requirements, cost of equipment and the time taken to perform dermoscopy were the most important barriers identified from the studies. However, for each barrier there were some papers that described it as a facilitator instead. Three papers performed cost-effective analyses of dermoscopy[24] and teledermoscopy,[26 41] and none found a significant cost-effective advantage. The main facilitators identified to the use of dermoscopy in primary care were reduced referrals, early detection of melanoma, and reduced patient and physician anxiety.

### Patient and PCP attitudes and acceptance of dermoscopy

Several papers assessed PCP attitudes to dermoscopy through questionnaires. Stratton and Loescher[42] found that nurse practitioners in the USA did not widely use dermatoscopes; however, they thought that dermoscopy would have a positive impact and would be willing to use mobile teledermoscopy if they received training. Morris et al[39 40] found that dermoscopy use among US physicians and doctors of osteopathic medicine was associated with seeing higher numbers of patients and with higher confidence in diagnosing skin lesions. Chappuis et al[38] found that dermoscopy use among French general practitioners (GPs) was associated with being older and male, and that only 8% of respondents had access to a dermatoscope. Livingstone and Solomon's[41] survey was the only one that assessed patient acceptability; they reported that 97% of patients from one general practice in Greater London were satisfied with the teledermoscopy service and 100% would recommend it.

### DISCUSSION
### Principal findings

Only a small number of studies have examined the use of dermoscopy or dermoscopy-related technologies in the primary care setting. These studies were all set in Europe, the USA and Australia, and due to their heterogeneous nature we were not able to synthesise the findings. Nevertheless, our review found that, with appropriate training, dermoscopy in primary care is more accurate than naked-eye examination, with improvements in sensitivity and specificity and number needed to excise. Furthermore, there was some evidence that teledermoscopy-based referral systems improve triage accuracy compared with paper-based or macroscopic image-based referral systems. The limited evidence did not show a significant cost-effectiveness benefit for either dermoscopy or teledermoscopy, although dermoscopy appears to lead to a reduction in unnecessary referrals and excisions. Importantly, the review also suggests that PCPs are receptive to incorporating dermoscopy into their routine practice, although they recognised ongoing implementation

**Table 3A** Summarised results of the RCTs and SITs

| Study | Summary | Control (C)/Intervention (I)(number of lesions) | Outcome measures | | | | |
|---|---|---|---|---|---|---|---|
| | | | Healthcare professional diagnosis | Expert/Histopathology diagnosis | Sens/Spec | PPV/NPV | Others |
| **RCTs and SITs** | | | | | | | |
| Argenziano et al[23] | RCT in primary care comparing PCPs using naked-eye observation (ABCD) with PCPs using dermoscopy (3-point checklist). | C (1325) | Non-susp=925 Susp=408 | 39 susp 23 malig 46 susp 30 malig | Sens 54.1% Spec 71.3% | PPV 11.3% NPV 95.8% | 2/6 MMs missed. |
| | | I (1203) | Non-susp=824 Susp=379 | 16 susp 6 malig 61 susp 33 malig | Sens 79.2% Spec 71.8% | PPV 16.1% NPV 98.1% | 1/6 MMs missed. |
| Koelink et al[24] | Cluster RCT in primary care comparing PCP diagnosis with naked-eye examination and dermoscopy examination. | C (230) | Non-susp=67 Referred=20 Biopsy/excision=135 | Correctly diagnosed: MMs 22.2% (2/9) Lesions 40.5% (90/222) | | | All lesions OR=1.51 Relative risk=1.25 MM OR=5.52 |
| | | I (207) | Non-susp=84 Referred=18 Biopsy/excision=92 | Correctly diagnosed: MMs 61.5% (8/13) Lesions 50.5% (98/194) 3 skin cancers incorrectly treated | | | |
| Menzies et al[14] | SIT using within-lesion controls in primary care assessing effect of dermoscopy and SDDI on management of suspicious PSLs by PCPs. | C (374) | 374 PSLs suspicious for referral ± excision | 42 malignant lesions 33 MM, 1 MM in situ | For MM: Naked-eye: sens 37.5%, spec 84.6%, PPV 20.7, NPV 92.7 Dermoscopy: sens 53.1%, spec 89.0%, PPV 34, NPV 94.7 +SDDI: sens 71.9%, spec 86.6%, PPV 36.4, NPV 96.6 | Overall: Increased sens for all malignancies (40%–67.5%, p=0.014) and MM (37.5%–71.9%, p=0.006) following intervention. Increased PPV for MM (20.7%–36.4%, p=0.055) and NPV for MM (92.7%–96.6%, p=0.041). | 56.4% reduction in suspicious PSLs excised/referred. 63.5% reduction in benign excised PSLs. 1MM in situ incorrectly managed in intervention group. Benign:MM ratio of excised/referred lesions 9.5:1 vs 3.7:1 (after dermoscopy) vs 3.5:1 (dermoscopy + SDDI) (p<0.0005). |
| | | IA (374) | After dermoscopy 110 referred, 192 SDDI, 72 observed for change | | | | |
| | | IB (192) | Of 192 SDDI, 46 referred/excised, 6 continued SDDI (2 subsequently referred, 4 observed), 140 observed (5 subsequently referred, 135 observed) | | | | |
| Rosendahl et al[43] | SIT using within-lesion controls. Comparison of PSL diagnosis of 'blinded observers' using macroscopic images, then dermoscopic images. | C (463) | Single best diagnosis matched HP diagnosis in 320 cases (69.1%) | 29 MMs, 72 BCCs, 5 SCCs NB: all PSLs excised | To achieve 80% spec, 70.5% sens | | AUC 0.83 (malignant neoplasms). AUC 0.71 (melanocytic lesions). |
| | | I (463) | Single best diagnosis matched HP diagnosis in 375 cases (80.1%) (p<0.001) | | To achieve 80% spec, 82.6% sens (NS) | | AUC 0.89 (malignant neoplasms) (p<0.001). AUC 0.76 (melanocytic lesions) (NS). |
| Ferrándiz et al[26] | RCT comparing DA and cost-effectiveness of clinical teleconsultations with clinical + dermoscopic teleconsultations from 5 primary care centres. | C (226) | 70.36% non-susp 45.14% referred for face-to-face evaluation | 2.77% MM, 11.54% non-MM skin cancer | Sens 86.57% Spec 72.33% | PPV 56.98 NPV 92.86 Accuracy index 79.20% | False negative rate 13.43%. False positive rate 22.16%. 69.71% decisions made with higher confidence. |
| | | I (228) | 73.24% non-susp, 20.18% referred for face-to-face evaluation (p<0.001) | 2.19% MM, 7.89% non-MM skin cancer | Sens 92.86% Spec 96.24% | PPV 84.38 NPV 98.17 Accuracy index 94.30% (p<0.001) | False negative rate 7.14%. False positive rate 3.76%. 78.07% decisions made with higher confidence (p=0.001). |

**Table 3B** Summarised results of the included non-RCT DA studies

| Study | Summary | Outcome measures | | | | | |
|---|---|---|---|---|---|---|---|
| | | Intervention or group (number of lesions) | Healthcare practitioner diagnosis | Expert review | Sens/Spec | PPV/NPV | Others |
| **Non-RCT DA studies** | | | | | | | |
| Ahmadi et al[27] | Retrospective cross-sectional study of medical files from 3 general practices. | (580) | 67 malignant, 75 premalignant, 399 non-suspicious, 39 unknown. 16.7% of patients referred. | 151 lesions confirmed by HP/dermatology: 37 BCC, 4 MMs, 1 lentigo maligna, 20 unknown. | PPV: benign lesions 85.7%, premalignant lesions 18.2%, malignant lesions 53.8%, BCCs 53.3%, melanoma 25%. | | Tools used: dermoscopy 8.4%, experienced PCP advice 1.4%, biopsy 1.9%, excision 10.3% |
| Bourne et al[28] | Sequential design DA study. 4 PCPs used new BLINCK dermoscopy algorithm on 50 lesion images, compared with same PCPs using 3-point checklist, Menzies and clinical assessment on same 50 images. | BLINCK (200) | Found 33/36 MMs. Biopsied 131/200 lesions. | Images of 50 lesions used: 1 invasive MM (0.52mm), 8 in situ. | Sens 90.8% Spec 50% | | DA 65.5% Number-needed-to-excise 6 |
| | | 3PCL (200) | Found 19/36 MMs. Biopsied 105/200 lesions. | | Sens 59.4% Spec 42.2% | | DA 48.3% Number-needed-to-excise 11 |
| | | Menzies (200) | Found 16/36 MMs. Biopsied 71/200 lesions. | | Sens 54.7% Spec 69% | | DA 63.9% Number-needed-to-excise 13 |
| | | Clinical (200) | Found 9/36 MMs. Biopsied 74/200 lesions. | | Sens 52.6% Spec 74.8% | | DA 65% Number-needed-to-excise 22 |
| Menzies et al[29] | Sequential design DA study comparing the performance of SolarScan with that of clinicians with varying dermatology experience on 78 images of PSLs. | Dermatologist (78) | 10.5 | | | Diagnosing MM: Sens 81%, spec 60%, PPV 30, NPV 94 Decision to excise: Sens 79%, spec 60%, PPV 29, NPV 93 | |
| | | PCP (78) | 8 | | | Diagnosing MM: Sens 62%, spec 63%, PPV 26, NPV 89 Decision to excise: Sens 62%, spec 61%, PPV 25, NPV 89 | |
| | | SolarScan (78) | 11.1 | | | Diagnosing MM: sens 85%, spec 65%, PPV 32, NPV 96 Decision to excise: sens 85%, spec 65, PPV 32, NPV 96 | SolarScan's sensitivity was higher than PCPs but NS. |
| Pagnanelli et al[30] | Sequential design DA study to assess if internet-based course suitable to teach dermoscopy to 16 clinicians with minimal dermoscopy experience looking at 20 images of PSLs. | Control: pretraining (20) | | 20 PSLs in test set: 6 MMs, 14 non-MMs. | Pattern analysis: Sens 67.7%, spec 76.3% ABCD: Sens 58%, spec 73.4% 7-point checklist: Sens 100%, spec 67.5% Menzies method: Sens 80.4%, spec 72% | | DA 49.8 k-intraobserver agreement 0.42 DA 38.8 k-intraobserver agreement 0.31 DA 60.2 k-intraobserver agreement 0.58 DA 53.5 k-intraobserver agreement 0.50 |
| | | Intervention: post-training (20) | | | Pattern analysis: Sens 82%, spec 78.5% ABCD: Sens 78.4%, spec 79.6% 7-point checklist: Sens 100%, spec 69.8% Menzies method: Sens 93.4%, spec 76% | | DA 60.1 k-intraobserver agreement 0.58 DA 56.6 k-intraobserver agreement 0.55 DA 64.5 k-intraobserver agreement 0.61 DA 62.8 k-intraobserver agreement 0.66 |
| Rogers et al[31] | Sequential design DA study examining performance of new TADA when used by 120 clinicians of various specialties looking at 50 PSL images. | (50) | 5641 lesion evaluations performed: 3034 malignant, 2607 non-suspicious. | 50 lesion images in test set, 23 benign, 27 malignant. Sens and spec calculated for malignant lesions. | Dermatologists: Sens 94.8%, spec 78.5% Non-dermatologists: Sens 93.7%, spec 72.1% >1year dermoscopy experience: Sens 95.4%, spec 77.3% <1year dermoscopy experience: Sens 91.3%, spec 74.2% | | |

Continued

**Table 3B** Continued

| Study | Summary | Intervention or group (number of lesions) | Outcome measures Healthcare practitioner diagnosis | Expert review | Sens/Spec | PPV/NPV | Others |
|---|---|---|---|---|---|---|---|
| Rogers et al[32] | Sequential design DA study comparing performance of new TADA with existing dermoscopy algorithms when used by 120 clinicians of various specialties. | (50) | No data for 3-point checklist and AC rule. TADA: 5646 lesions evaluated. 2056 deemed non-suspicious (1891 true negatives (92.0%), 165 false negatives (8.0%)). 3590 deemed malignant (2871 true positives (80.0%), 719 false positives (20.0%)). | 50 lesion images in test set, 23 benign, 27 malignant. Sens and spec based on 40 non-PSLs. | TADA: Sens 94%, spec 75.5%; AC: Sens 88.6%, spec 78.7%; 3-point checklist: Sens 71.9%, spec 81.4%; Untrained (using TADA): Sens 93.6%, spec 69%; Trained (using TADA): Sens 95.4%, spec 73.2% | | Sens for MM with TADA 94%. Spec for untrained clinicians for benign PSLs using TADA 76%–94% (beginners can be quickly trained to identify benign lesions). |
| Rosendahl et al[25] | Prospective cohort study using SCARD to assess impact of dermoscopy use and subspecialisation on MM diagnosis by PCPs. | Dedicated skin cancer practitioners | Number of lesions seen not recorded on SCARD. 11 992 lesions referred. | 11.7% were MM. | | | MM number needed to treat: 8.5 |
| | | PCP with special interest in skin cancer | 1942 lesions referred. | 10.6% MM. | | | MM number needed to treat: 9.4 |
| | | PCPs | 1942 lesions referred. | 5.9% MM. | | | MM number needed to treat: 17.0 |
| | | High dermoscopy use | 17 917 lesions referred. | 11.2% MM. | | | MM number needed to treat: 8.9 |
| | | Medium use | 2657 lesions referred. | 9.1% MM. | | | MM number needed to treat: 10.9 |
| | | Low use | 1093 lesions referred. | 6.9% MM. | | | MM number needed to treat: 14.6 (p<0.0001, but NS when adjusted for subspecialisation) |
| Secker et al[44] | Sequential design DA study comparing performance of 293 PCPs in diagnosing PSLs before and after a training intervention. | Pretest (clinical images, no education) (20) | | 20 PSL images in test set: 3 MM, 2 BCC, 15 benign. | For MM: Sens 0.49%, spec 0.75% | | % correct treatment: (1) malignant 85.87, (2) naevi 92.83, (3) benign 9.56 |
| | | Post-test (clinical images with education) (20) | | | Sens 0.50%, spec 0.77% | | (1) malignant 84.03, (2) naevi 95.61, (3) benign 7.81 |
| | | Integrated post-test (clinical and dermoscopic images with education) (20) | | | Sens 0.66%, spec 0.70% | | (1) malignant 91.74, (2) naevi 92.35, (3) benign 29.35 |
| | | Overall | | | Training improved sens and spec for all except pigmented naevi. Training improved DA for all PSLs except naevi. | | Increase in correct treatments for benign lesions, reduced unnecessary referrals and excisions. |
| Westerhoff et al[33] | Intervention study assessing performance of 74 PCPs with no dermoscopy experience in diagnosing 100 PSLs using macroscopic images ± dermoscopy images before and after an educational intervention. | Control: no education (100) | | 100 images of lesions used: 50 invasive MMs and 50 non-melanomas. | Macroscopic images: Pretest: Sens 50.6%, spec 55.2% Post-test: Sens 53.7%, spec 51.5% +Dermoscopic images: Pretest: Sens 52.9%, spec 58.1% Post-test: Sens 54.8%, spec 55.8% | | Significant improvement with training between pretest (54.6%) and post-test (62.7%) (p=0.007) on macro images. Diagnosis of MM with dermoscopy significantly better (75.9%) than macro images (62.7%)(p=0.000007). No significant difference adding dermoscopic images in diagnosing non-MM PSLs. |
| | | Intervention: with education (100) | | | Macroscopic images: Pretest: Sens 54.6%, spec 53.0% Post-test: Sens 62.7%, spec 53.6% +Dermoscopic images: Pretest: Sens 57.8%, spec 55.5% Post-test: Sens 75.9%, spec 57.8% | | |

Continued

**Table 3B** Continued

| Study | Summary | Intervention or group (number of lesions) | Outcome measures | | | | |
|---|---|---|---|---|---|---|---|
| | | | Healthcare practitioner diagnosis | Expert review | Sens/Spec | PPV/NPV | Others |
| Börve et al[34] | Case-control study. Smartphone TDS system in 20 primary healthcare centres compared with traditional, paper-based referral system from other primary healthcare centres. | Control: paper-based referrals (746) | 746 suspicious lesions referred. | 323 malignant (13 MM, 7 MM in situ, 22 SCCs, 115 BCCs, 164 AKs), 423 benign. | | | 3/4 invasive MMs given medium/low priority, 3/5 MMs in situ given low priority. Mean response time 5 days (range 0–82 days). Patients received primary treatment on single face-to-face visit in 82.2% of cases. |
| | | Intervention: smartphone TDS referrals (816) | 816 suspicious lesions referred. 346 (42%) non-suspicious, final diagnosis benign for 343 (3 malignant lesions missed were AKs). 196 deemed malignant, 146 (74%) also malignant after dermatology/HP. | 229 malignant (19 MM, 16 MM in situ, 24 SCCs, 109 BCCs, 61 AKs), 587 benign. | | | All invasive MMs prioritised correctly (high), all MM in situ at least medium priority. 22.6% more referrals given low priority. Mean response time 109 min (range 2 min to 46 hours). Waiting time for surgical treatment for MM significantly shorter (p<0.0001). Patients received primary treatment on single face-to-face visit in 93.4% of cases. |
| Grimaldi et al[35] | Sequential design DA study assessing PCP diagnosis of suspicious PSLs before and after dermoscopic evaluation and accuracy of teledermatology triage system. | PCP clinical (235) | 167 lesions non-suspicious, 68 suspicious. | 16 malignant (5 MMs), 219 benign. | Dermoscopy by PCPs and then experts led to 76.5% reduction in number of surgical procedures (68 to 16). | | PCP clinical vs PCP dermoscopy: p<0.001, OR 0.345731 |
| | | PCP dermoscopy (235) | 206 non-suspicious, 29 suspicious (dermoscopy-corrected diagnosis in 57.3% of cases, only 1 false negative). | | | | PCP clinical vs expert dermoscopy: p<0.001, OR 0.179425 |
| | | TDS (235) | 219 non-suspicious, 16 suspicious (1 false negative from 206 benign lesions). | | | | PCP dermoscopy vs expert dermoscopy: p<0.05, OR 0.518973 |
| Livingstone and Solomon[41] | Prospective case series to assess cost-effectiveness, accuracy and patient satisfaction of a TDS system for non-malignant PSLs in a primary care practice. | (248) | 248 patients that PCP would have been referred routinely to dermatology referred to TDS service. 102 needed face-to-face dermatology review. 146 advised on treatment. 3 lesions possibly malignant so referred 2-week-wait pathway. | 0/3 possibly malignant lesions were malignant at face-to-face review. None of other 245 lesions were malignant after review or follow-up. | | | Waiting time for images to be taken (weeks): 0–1=27%, 1–2=45%, 2–3=7%, 3–4=0%, 4–5=7%, 5–6=7%. Waiting time for results (weeks): 0–1=14%, 1–2=61%, 2–3=14%, 3–4=7%. 129 patients returned patient satisfaction questionnaires. 100% said TDS service was explained, 100% would recommend TDS. |
| Moreno-Ramirez et al[36] | Sequential design DA study to assess if teledermatology with dermoscopy images would improve the current teledermatology-based triage system in referrals from a primary care centre. | Control: teledermatology referrals (61) | 4 BCCs, 1 MM, 0 dysplastic naevus, 56 benign. Referral rates 47.5% (29). Rate of referral of true positive results 17.2% (5 true positives/29 referrals). False positives 58.7% (17/29 referrals). | HP: 2 BCCs, 1 MM, 1 dysplastic naevus, 57 benign. | Sens 1 (as 0 false negative), spec 0.65, false positive rate 0.35 | | Clinical picture quality excellent 41%, poor 3.3%. Average diagnostic confidence 4.14/5. Agreement with histology 0.91. |
| | | Intervention: TDS referrals (61) | 2 BCCs, 1 MM, 1 dysplastic naevus, 54 benign. Referral rates 39.3% (24) (p<0.05). Rate of referral of true positive results 20.8% (5 true positives/24 referrals) (p<0.05). False positives 41.7% (10/24) (p<0.05). | | Sens 1 (as 0 false negatives), spec 0.78 (p<0.05), false positive rate 0.22 (p<0.05) | | Dermoscopic picture quality excellent 63.9%, poor 6.6%. Average diagnostic confidence 4.75/5 (p<0.05). Agreement with histology 0.94. |
| van der Heijden et al[37] | Cohort study assessing accuracy and reliability of TDS diagnosis with images taken by PCPs compared with diagnosis at face-to-face consultations for same lesions. | Control: face-to-face assessment (76) | All 108 lesions also referred for face-to-face assessment. 76 lesions seen face-to-face by dermatology. 32 not seen as did not attend, moved away, GP did excision. | | | | Agreement face-to-face vs HP diagnosis k=0.90 (almost perfect), diagnostic agreement k=0.56–0.78 (substantial), management agreement k=0.31–0.38 (fair). |
| | | | | | | | Agreement TDS vs face-to-face diagnosis k=0.55–0.73 (moderate–substantial), TDS vs face-to-face management k=0.19–0.29 (fair). Image quality: 36% bad, 36% good. TDS consultations with good image quality had better agreement, TDS vs face-to-face diagnosis (k=053–0.77, substantial), and TDS vs face-to-face management (k=0.34–0.47, fair-moderate). |
| | | Intervention: TDS referrals (108) | 108 lesions referred via TDS. | HP diagnosis for 36 lesions (33%). 2 MMs and 5 non-melanoma skin cancers. | | | Agreement TDS vs HP diagnosis k=0.41–0.63 (moderate). TDS consultations with good image quality had better agreement of TDS vs HP diagnosis (k=0.53–1.0, moderate–almost perfect). |

**Table 3C** Summarised results of the included survey-based studies

| Study | Summary | Population studied (N) | Outcomes |
|---|---|---|---|
| Survey-based studies | | | |
| Chappuis et al[38] | Survey of PCPs in 3 regions of France. | PCPs (425) | Among dermoscopy users, 21 (54%) had no training, 8 (21%) trained via books, 5 (13%) trained with dermatologist, 2 (5%) trained online. Lower referral rates in dermoscopy group. Male PCPs significantly more likely to use a dermatoscope (p=0.001). PCPs >50 significantly more likely to use a dermoscopy (p<0.001). 30 (8%) had dermatoscope available, 16 (52%) used it >1×/week. |
| Morris et al[39] | Descriptive cross-sectional survey of US physicians (medical doctors and doctors of osteopathy) to examine dermoscopy use and barriers. | Family physicians (705) | Confidence recognising malignant lesions: not confident=2.1%, a little confident=18.8%, moderate=21.9%, confident=47.7%, very confident=9.4%. Currently using a dermatoscope associated with seeing >400 patients/month and >60 years. Number of patients per month with suspicious lesions: <1.5 lesions=12.2%, 1.5–4.99 lesions=19.6%, 5–9.99 lesions=19.2%, 10–19.99 lesions=23.1%, >20 lesions=26%. Used a dermatoscope=19.5%. Currently use a dermatoscope=8.3%. Intention to start using in 12 months. |
| Morris et al[40] | Same study but including all clinicians. | Physicians (1466) | 211 (14.6%) had used dermoscopy, 87 (6.0% of 1445) currently using, 656 (51.8%) intended to use in next 12 months. Use of and intention to use dermoscopy were associated with graduating recently, being a family physician, seeing a higher number of patients with cancer and being more confident differentiating malignant and benign skin lesions. |
| Stratton and Loescher[42] | Online survey, acceptance of mobile teledermoscopy by nurse PCPs in Arizona, USA. | Nurse practitioners (62) | Practitioners 40–60 years and been in practice for 1–15 years scored higher for intention to use mobile teledermoscopy. Few nurse practitioners used mobile teledermoscopy. They scored highly for perceiving that mobile teledermoscopy would have a positive impact on their practice, they would find it interesting to use, they could easily learn mobile teledermoscopy, it would help with rapid diagnosis of skin cancer and would improve the diagnosis of their patients, and they would use mobile teledermoscopy if they received training. |

ABCD, Area, Border, Colour, Diameter; AC Rule, Asymmetry, Colour variation; AK, actinic keratosis; AUC, area under the curve; BCC, basal cell carcinoma; BLINCK, Benign, Lonely, Irregular, Nervous, Change, Known clues; DA, diagnostic accuracy; GP, general practitioner; HP, histopathology; MM, malignant melanoma; NPV, negative predictive value; NS, not significant; OR, Odds Ratio; 3PCL, 3-point checklist;PCP, primary care physician; PPV, positive predictive value; PSL, pigmented skin lesion; RCT, randomised controlled trial; SCARD, Skin Cancer Audit Research Database; SCC, squamous cell carcinoma; SDDI, short-term sequential digital dermoscopy imaging; SIT, sequential intervention trial; TADA, triage amalgamated dermoscopic algorithm; TDS, teledermoscopy; malig, malignant; sens, sensitivity; spec, specificity; susp, suspicious.

barriers, particularly around training, time requirements and technology costs.

## Comparison with other studies

Our review suggests that dermoscopy in primary care is more accurate than naked-eye examination, supporting the findings from a previous review of dermoscopy for melanoma detection specifically in primary care published in 2012.[45] A recently published Cochrane review of dermoscopy for the diagnosis of melanoma has also concluded that, although data to support dermoscopy use in primary care are limited, 'it may assist in triaging suspicious lesions for urgent referral when employed by suitably trained clinicians'.[19] Our review also suggests that training PCPs in dermoscopy improves diagnostic accuracy. Again, this finding is supported by the recent Cochrane review which also suggests that 'formal algorithms may be of most use for dermoscopy training purposes and for less expert observers, however reliable data comparing approaches using dermoscopy in-person are lacking'.[19] Previous reviews have shown that using dermoscopy without training was no more accurate than naked-eye examination alone.[13 19] However, we were not able to identify the optimal length of training needed to train PCPs to use dermoscopy accurately, although studies of the effect of training on dermatologist diagnostic performance have shown improvement after between 2 days (6 hours of training per day)[46] and 10 weeks (comprising 6 workshops of 4–6 hours).[47] PCPs are likely to need short training courses, preferably with regular updates, as one of the few RCTs examining the impact of dermoscopy on the management of pigmented lesions in primary care reported a high dropout rate of GPs from the 20 hours of online training required for that study.[14]

It is important to note that the performance of dermoscopy in specialist clinics is not directly translatable as evidence for the performance of dermoscopy in primary care settings. A spectrum effect or spectrum bias is often observed when tests developed in one population are then used on another population. For example, the secondary care population is a referred population and has a higher prevalence of the condition being tested than primary care populations. This means that a diagnostic test, such as dermoscopy, will perform differently in the primary care population with the lower prevalence of the condition, compared with the secondary care population.[48] The direction of effect is not consistent across tests and conditions; hence, to establish the performance of tests among the non-referred population in primary care, they need to be evaluated in a primary care population. This review has therefore aimed to examine existing evidence for dermoscopy use in primary care settings.

Our review suggests that a range of PCPs, including nurse practitioners in the USA, and PCPs in the USA and France, hold positive views about incorporating dermoscopy into their routine practice. Evidence from Australia supports these views and demonstrates that

**Table 4** Barriers and facilitators to implementation of dermoscopy and teledermoscopy

| Aspect | Quoted as barrier in: | Type of study | Quoted as facilitator in: | Type of study |
|---|---|---|---|---|
| Training requirements | Chappuis et al[38] | Survey | Pagnanelli et al[30] | DA study |
| | Morris et al[40] | Survey | | |
| | van der Heijden et al[37] | Cohort study | | |
| Cost* | Chappuis et al[38] | Survey | Koelink et al[24]* | RCT |
| | Morris et al[39] | Survey | Rosendahl et al[25] | Cohort study |
| | Moreno-Ramirez et al[36] | DA study | Ferrándiz et al[26]* | RCT |
| | | | Livingstone and Solomon[41]* | Case series |
| Time consumption | Chappuis et al[38] | Survey | Börve et al[34] | Case–control |
| | Moreno-Ramirez et al[36] | DA study | | |
| | van der Heijden et al[37] | Cohort study | | |
| Reimbursement for offering dermoscopy services (in USA) | Morris et al[39] | Survey | | |
| Equipment issues | van der Heijden et al[37] | Cohort study | Börve et al[34] | Case–control |
| | | | Moreno-Ramirez et al[36] | DA study |
| Reduced referrals | | | Chappuis et al[38] | Survey |
| | | | Koelink et al[24] | RCT |
| | | | Börve et al[34] | DA study |
| | | | Moreno-Ramirez et al[36] | DA study |
| Early detection of melanoma | | | Chappuis et al[38] | Survey |
| Reduced patient anxiety | | | Chappuis et al[38] | Survey |
| Reduced physician anxiety | | | Chappuis et al[38] | Survey |
| | | | Moreno-Ramirez et al[36] | DA study |
| | | | Menzies et al[14] | DA study |

*Based on studies where a cost-effective analysis was undertaken.
DA, diagnostic accuracy; RCT, randomised controlled trial.

a wide range of PCPs are able to incorporate dermoscopy into their routine clinical practice.[16] Only a small number of cost-effectiveness studies met our review criteria. They all assessed dermoscopy and teledermoscopy from a healthcare perspective, and only reported on short-term costs resulting from dermoscopy or non-dermoscopy approaches. None reported a significant cost-effectiveness benefit for dermoscopy[24] or teledermoscopy,[26] in the primary care setting, although they recommended the technologies as potentially useful tools. An English RCT of another diagnostic aid (MoleMate, incorporating SIAscopy) in primary care[49] also reported equivocal findings on cost-effectiveness, as the device, similar in accuracy to systematic application of the 7-point checklist, resulted in increased referrals from primary care.[49 50]

Interestingly, no papers reporting the use of dermoscopy smartphone applications ('apps') for automated diagnosis of melanoma or skin cancers in the primary care setting met the review inclusion criteria. Kassianos et al[51] reviewed 39 smartphone applications and found little evidence of clinical or research-based input into the design or evaluation of these apps. A recent editorial in *The Lancet Oncology* supported this finding,[52] and urged caution with early adoption of new technologies that are often poorly designed and untested, stressing the need to ensure that these technologies are appropriate,

cost-effective and do not compromise patient safety. Recent studies have tested the application of artificial intelligence, neural networks and machine learning to the diagnosis of skin lesions; however, they have not yet been assessed in primary care settings.

### Strengths, limitations and future research

Our review examines the evidence for dermoscopy use in the primary care setting. It builds on the recent Cochrane review which explicitly reviewed evidence only about diagnostic accuracy of dermoscopy, with and without naked-eye examination, and describes this in specialist and generalist settings.[19] We therefore included studies with a range of methods, surveys and qualitative studies, as well as RCTs and diagnostic accuracy studies, but still only identified a relatively small number of publications. Unfortunately, we were unable to perform a meta-analysis due to the heterogeneity in study designs, settings, populations and outcomes. All the studies are from high-income countries and therefore may be less generalisable to other countries with different healthcare systems.

### CONCLUSIONS

Despite the limited evidence, this review provides moderate support for the use of dermoscopy in primary

care, with the weight of the available evidence pointing to a benefit in diagnostic accuracy for managing suspicious skin lesions. Dermoscopy is acceptable to PCPs, so it could help them triage suspicious lesions for urgent referral or reassurance. However, it will be important to establish further evidence on minimum for training to reach competence, as well as the cost-effectiveness and patient acceptability of implementing dermoscopy in primary care.

**Acknowledgements** The authors thank Isla Kuhn, Reader Services Librarian, University of Cambridge Medical Library, for her help in developing the search strategy, and Margaret Johnson, a patient advocate, who provided regular comments on the study from its conception.

**Contributors** OJ developed the protocol, completed the search, screened the articles for inclusion, extracted the data, synthesised the findings, interpreted the results and drafted the manuscript. LJ screened the articles for inclusion, extracted the data and critically revised the manuscript. MvM screened the articles for inclusion, extracted the data and critically revised the manuscript. SH screened the articles for inclusion and critically revised the manuscript. NB developed the protocol and critically revised the manuscript. PH developed the protocol and critically revised the manuscript. JE developed the protocol, interpreted the results and critically revised the manuscript. FMW developed the protocol, synthesised the findings, interpreted the results and critically revised the manuscript. All authors approved the final version.

**Funding** This research arises from the CanTest Collaborative, which is funded by Cancer Research UK (C8640/A23385), of which FW is Director and JE is Associate Director. This work was also supported by FMW's Clinician Scientist Award (RG 68235) from the National Institute for Health Research (NIHR). The views expressed in this publication are those of the authors and not necessarily those of the National Health Service, the NIHR or the Department of Health. The funding sources had no role in the study design, data collection, data analysis, data interpretation, writing of the report or in the decision to submit for publication.

**Competing interests** PH is Clinical Advisor for Skin Cancer to Check4Cancer, a company that offers teledermatoscopic analysis of pigmented skin lesions (https://www.check4cancer.com), and is a medical advisor to MedX, the Canadian company with the IP for the SIAscope (https://medxhealth.com/default.aspx).

**Patient consent for publication** Not required.

**Provenance and peer review** Not commissioned; externally peer reviewed.

**Data availability statement** Data are available upon reasonable request.

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
