## [Reviewer comments · BMJ Open]

ARTICLE DETAILS

TITLE (PROVISIONAL)	Dermoscopy for melanoma detection and triage in primary care: A systematic review
AUTHORS	Jones, Owain; Jurascheck, Leo; van Melle, Marije; Hickman, Sarah; Burrows, Nigel; Hall, Per; Emery, Jon; Walter, Fiona

VERSION 1 - REVIEW

REVIEWER	John Paoli Dept. of Dermatology and Venereology, Institute of Clinical Sciences, Sahlgrenska Academy, University of Gothenburg, Gothenburg, Sweden I am a small shareholder in the teledermoscopy company iDoc24 AB (Sweden).
REVIEW RETURNED	05-Nov-2018

GENERAL COMMENTS	The authors present an interesting systematic review of 23 studies carefully selected according to the PRISMA guidelines on whether or not dermoscopy and dermoscopy-related technologies can be used accurately and effectively to triage suspicious skin lesions in primary care. Although a meta-analysis was not carried out due to the heterogeneous nature of the studies, the results and summary in this review are still clinically relevant. I recommend that this manuscript be published following a few minor revisions (see list below). - Table 1 appears after Table 2 in the manuscript.- Page 24, line 27. It should be clarified that you are referring to dermoscopy smartphone apps for automated diagnosis of melanoma or skin cancer.
--

REVIEWER	Prof Isabelle Tromme Melanoma Clinic Catholic University of Louvain St Luc Hospital Brussels Belgium
REVIEW RETURNED	25-Nov-2018

GENERAL COMMENTS	Well written and very detailed paper. In my personal opinion, the tables are too detailed. This gives them quite difficult to "digest". On the other hand, their summary in
--

	the text is sometimes too short and not detailed enough, especially in the "results" part. Introduction. First §: please add the difference between important increasing of incidence and slow increasing /stabilisation of mortality Third §: please consider the following paper: Man against machine.... Haenssle et al Annals of Oncology 2018 instead or in addition to your ref 17 Results Tables: the authors should help the reader by linking the study (expressed by the first author's name and the year) with the number in the references Table 1: what is 2F2? The comments (in the text) about Table 1 only refer to the first part of the table (dermoscopy). This should be mentioned. The last sentence (Overall, consultations) is unclear to me) Table 2: JBI score needs an explanation, at least a reference, and some comments in the main text Discussion: please explain what is spectrum effect. The sentences : Spectrum..... care setting are not clear to me.
--	--

REVIEWER	Celia Álvarez-Bueno Health and Social Research Center
REVIEW RETURNED	22-Dec-2018

GENERAL COMMENTS	Thank you for the opportunity to review this interesting paper on the use of dermoscopy for melanoma detection. It provides a comprehensive review on an important health related issue. I would like to offer some suggestions that I hope will be helpful. Introduction section seems to be focused on UK data, while a systematic review tries to provide a worldwide comprehensive synthesis of the evidence. I encourage the authors to provide a more general perspective. Last paragraph of introduction section is not clear. Please, it could be helpful to clearly state the aims of the review. It should be also considered, that not all study designs are useful to answer the same research question, which should be clarified in methods section. Methods. In page 5, line 18-20 the authors state "We also considered published evidence from international healthcare systems and whether it could be interpreted and applied to the UK ...". It is not clear the rationale of this decision. Inclusion and exclusion criteria seem to contradict themselves. Search strategy was finished one year ago, it should be updated.
---

VERSION 1 – AUTHOR RESPONSE

Reviewer 1: John Paoli

The authors present an interesting systematic review of 23 studies carefully selected according to the PRISMA guidelines on whether or not dermoscopy and dermoscopy-related technologies can be used accurately and effectively to triage suspicious skin lesions in primary care. Although a meta-analysis was not carried out due to the heterogeneous nature of the studies, the results and summary in this review are still clinically relevant. I recommend that this manuscript be published following a few minor revisions (see list below).

- Table 1 appears after Table 2 in the manuscript.

Thank you for this observation, the citations have been corrected so the tables now appear in the correct order.

- Page 24, line 27. It should be clarified that you are referring to dermoscopy smartphone apps for automated diagnosis of melanoma or skin cancer.

Thank you for this suggestion, the relevant sentence has been modified to clarify the technology that is being referred to.

Reviewer 2: Prof Isabelle Tromme

Well written and very detailed paper.

In my personal opinion, the tables are too detailed. This gives them quite difficult to "digest". On the other hand, their summary in the text is sometimes too short and not detailed enough, especially in the "results" part.

We appreciate the reviewer's opinion, but, as the other 2 reviewers did not express similar views, we would prefer to retain the tables, and have added to the text in the Results section.

Introduction.

First §: please add the difference between important increasing of incidence and slow increasing /stabilisation of mortality

Many thanks for this suggestion. Whilst it is an important point, and given the limited word count for the article, we do not feel that there is room to go in depth on this point.

Third §: please consider the following paper: Man against machine.... Haenssle et al Annals of Oncology 2018 instead or in addition to your ref 17 Results

Many thanks for bringing this citation to our attention. The results from this paper indeed supersede the one cited in our article and have replaced the previous reference.

Tables: the authors should help the reader by linking the study (expressed by the first author's name and the year) with the number in the references

Many thanks for this suggestion, all studies have now been referenced.

Table 1:

What is 2F2?

F2F was used as an abbreviation for 'face-to-face'. The text has been altered to remove this, many thanks for bringing this to our attention.

The comments (in the text) about Table 1 only refer to the first part of the table (dermoscopy). This should be mentioned. The last sentence (Overall, consultations) is unclear to me)

Many thanks for this suggestion. We have rewritten this sentence to expand on the results and to clarify where we were referring to the dermoscopy studies and where we were referring to the teledermoscopy studies.

Table 2: JBI score needs an explanation, at least a reference, and some comments in the main text

The Joanna Briggs Institute (JBI) critical appraisal tools are now referenced in the table. The JBI tools are explained in the last paragraph of the methods and the JBI outcomes are discussed briefly in the results paragraph that pertains to table two.

Discussion: please explain what is spectrum effect.

The sentences : Spectrum..... care setting are not clear to me.

Many thanks for this suggestion: this sentence has been re-written to clarify what is meant by a spectrum effect. An excellent article reviewing the spectrum effect has also been referenced for the readers' information.

Reviewer 3: Celia Álvarez-Bueno

Thank you for the opportunity to review this interesting paper on the use of dermoscopy for melanoma detection. It provides a comprehensive review on an important health related issue. I would like to offer some suggestions that I hope will be helpful.

We are very grateful for these positive comments and suggestions.

Introduction section seems to be focused on UK data, while a systematic review tries to provide a worldwide comprehensive synthesis of the evidence. I encourage the authors to provide a more general perspective.

Many thanks for this suggestion: the statistics and focus of the introduction have been amended to emphasise a worldwide perspective.

Last paragraph of introduction section is not clear. Please, it could be helpful to clearly state the aims of the review.

Many thanks for this suggestion: we have amended this paragraph to clarify its meaning. We have retained the sentence, "Our systematic review has a broader aim, focussing on the first presentation of suspicious skin lesions in primary care and whether dermoscopy and dermoscopy-related technologies, with suitable training, can be used accurately and effectively to triage suspicious skin lesions at this point in the healthcare pathway" as we believe that it clearly states the aim of the review.

It should be also considered, that not all study designs are useful to answer the same research question, which should be clarify in methods section.

Many thanks for this suggestion: this was something we considered, and we have altered the relevant paragraph in the methods section to better explain this point.

Methods. In page 5, line 18-20 the authors state "We also considered published evidence from international healthcare systems and whether it could be interpreted and applied to the UK ...". It is not clear the rationale of this decision.

The organisation of healthcare systems around the world is very variable. In order to increase the yield of papers from our search we considered published evidence from healthcare systems where the distinction between primary and secondary care was not applicable, and whether this evidence could be applied to primary care healthcare models. This sentence has now been modified and hopefully clarified.

Inclusion and exclusion criteria seem to contradict themselves.

We believe that here you are referring to the sentence regarding "studies of secondary care physicians who were not trained in dermoscopy were also included". We have added more detail to this sentence to demonstrate why these criteria do not contradict themselves. The rationale behind this sentence was that secondary care physicians with no training in dermoscopy are roughly equivalent to primary care physicians with no training. There were few papers which included secondary care physicians with no dermoscopy training, they were all carefully analysed and discussed to assess whether there was data relevant to the research question which could be extracted from them. Where relevant data was found these studies were included.

Search strategy was finished one year ago, it should be updated.

We are not aware of any significant additions to the literature since this date on the use of dermoscopy in primary care settings for the triage of melanoma, and neither of the other 2 reviewers felt this was necessary. Nevertheless, if the editor felt it was required, we would be happy to consider updating the search.